# Effect of Pd-Doping Concentrations on the Photocatalytic, Photoelectrochemical, and Photoantibacterial Properties of CeO₂

**Shaidatul Najihah Matussin** [1], **Fazlurrahman Khan** [2,3], **Mohammad Hilni Harunsani** [1], **Young-Mog Kim** [2,3,4] and **Mohammad Mansoob Khan** [1,*]

1. Chemical Sciences, Faculty of Science, Universiti Brunei Darussalam, Jalan Tungku Link, Gadong BE 1410, Brunei
2. Marine Integrated Biomedical Technology Center, The National Key Research Institutes in Universities, Pukyong National University, Busan 48513, Republic of Korea
3. Research Center for Marine Integrated Bionics Technology, Pukyong National University, Busan 48513, Republic of Korea
4. Department of Food Science and Technology, Pukyong National University, Busan 48513, Republic of Korea
* Correspondence: mmansoobkhan@yahoo.com or mansoob.khan@ubd.edu.bn

**Abstract:** Cerium oxide (CeO₂) can exhibit good photocatalytic and photoantibacterial activities. However, its light-harvesting property is rather limited due to its large band gap. In order to boost these properties, doping with metal ions can improve light absorption and charge mobility. In this report, CeO₂ and palladium−doped CeO₂ (Pd−CeO₂) NPs were synthesized via the microwave-assisted synthesis method. The structural, optical, and morphological studies of CeO₂ and Pd−CeO₂ NPs were carried out using various techniques. Mixed phases of CeO₂/Ce₂O₃ were observed in pure CeO₂ (S−CeO₂) and Pd−CeO₂. However, the Ce₂O₃ phase gradually disappeared upon doping with a higher percentage of Pd. Almost spherical particles were observed with average sizes between 6 and 13 nm. It was found that the incorporation of Pd reduced the particle size. Moreover, band gap energies of S−CeO₂ and Pd−CeO₂ NPs were reduced from 2.56 to 2.27 eV, and the PL intensities were also quenched with more Pd doping. The shifts in the conduction band and valence band were found to cause the reduction in the band gap energies of S−CeO₂ and Pd−CeO₂ NPs. In the case of photocatalytic degradation of methylene blue, photoelectrochemical, and photoantibacterial activities, Pd−CeO₂ NPs showed enhanced activities under visible light irradiation. Therefore, Pd−CeO₂ NPs have been shown to be a visible-light active material.

**Keywords:** cerium oxide; CeO₂; Pd−doped CeO₂; noble metal; oxygen vacancies; MB degradation; antibacterial studies



## 1. Introduction

Antibiotics have been used to treat bacterial infections that are major causes of chronic infections and mortality. However, studies found that the widespread use of antibiotics has led to the emergence of multidrug-resistant bacterial strains [1]. In general, the major groups of antibiotics that are currently in use have three bacterial targets, namely: cell wall synthesis, translational machinery, and DNA replication machinery [2]. However, bacterial resistance can develop at each mode of action. Nanoparticles (NPs) have demonstrated broad-spectrum antibacterial properties against Gram-positive and Gram-negative bacteria. Studies have revealed that NPs do not show antibiotic resistance mechanisms because NPs contact directly with the bacterial cell wall, which would be less prone to promoting resistance in bacteria. Therefore, NPs can be potentially used as an alternative antibiotic. According to research, the major processes underlying the antibacterial effects of NPs are disruption of the bacterial cell membrane, generation of ROS, penetration of bacterial cell

membrane, and induction of intracellular antibacterial effects, including interactions with DNA and proteins [2,3].

Organic dyes, which generally consist of non-biodegradable, highly poisonous, and colored pigments, have been reported to be present and widely spread in industrial wastewater originating from the paper, textile, and apparel industries [4,5]. Even at low concentrations, dyes can be clearly seen and pollute various aquatic environments [6]. These dye-polluted effluents are harmful to living organisms [7]. Dyes such as methylene blue (MB) can also cause serious threats to humans, for example, abdominal disorders, respiratory problems, and digestive and mental disorders [8,9]. Hence, the removal of dyes from wastewater is important. Apart from that, redox reactions have shown promising water treatment activity as the reaction products can be conveniently separated and removed [10]. Owing to this, researchers have applied electrochemical systems to diverse functional applications, including photovoltaics [11], fuel cells [12], supercapacitors [13], as well as sensors [14]. It is reported that heterostructure exhibits interesting electrochemical properties owing to the possibility of a dual charge storage contribution from both materials [15]. Nevertheless, materials with high porosity and large surface areas are highly suitable for electrochemical applications [16].

Metal oxides such as $TiO_2$ [17], $ZnO$ [18], $Fe_2O_3$ [19], $CeO_2$ [20], etc., have shown remarkable activities in biological and photocatalytic activities. Among these, $CeO_2$ has shown extensive industrial applications in medicine, catalysis, solid oxide fuel cells, luminescence, optical, and sensor technologies [21–25]. Despite being a highly abundant material, its redox ability to change between $Ce^{3+} \leftrightarrow Ce^{4+}$ has also attracted significant attention. $CeO_2$ has been synthesized using hydrothermal, sol-gel, precipitation, microwave, and green synthesis methods where multiple morphologies such as cube, rod, spherical, and flower-like particles have been produced [26–30]. In catalytic applications, the activity of $CeO_2$ primarily arises from the reduction of $Ce^{4+}$ to $Ce^{3+}$ and the formation of oxygen vacancies [31,32]. The modification in shape and size results in the formation of surface defects such as $V_o$, which endows it with the ability to exist in either the $Ce^{3+}$ or $Ce^{4+}$ state on the surface of the particle [33,34].

In addition, dopants with different ionic states incorporated into $CeO_2$ can generate more structural defects to gain charge neutrality, enhancing physical properties and biocompatibility. Many studies have emphasized a strong synergy between $CeO_2$ and noble metals, which can significantly enhance catalytic activities [35]. Among the noble metals, palladium is increasingly used in electrical equipment, dental materials, implanted medical devices and automobiles as catalysts [36]. The incorporation of $Pd^{2+}$ into the $CeO_2$ crystal lattice has shown better catalytic activity than the palladium-supported $CeO_2$ catalyst [37]. It is believed that the unique properties of Pd-doped $CeO_2$ depend on the active components of palladium and the interaction between palladium and $CeO_2$ [38].

Therefore, in this study, $CeO_2$ (S−$CeO_2$) and Pd−doped $CeO_2$ (Pd−$CeO_2$) NPs were synthesized using a microwave-assisted synthesis method. To the best of the authors' knowledge, a study on Pd−$CeO_2$ with mixed phases of $CeO_2$ and $Ce_2O_3$ has not yet been explored. Moreover, studies on the photocatalytic degradation of methylene blue (MB) were conducted using synthesized-$CeO_2$ (S−$CeO_2$) and (0.5, 1, 3 and 5%) Pd−$CeO_2$ under visible light irradiation for 5 h. The photoelectrochemical studies, namely, linear sweep voltammetry (LSV) and electrochemical impedance spectroscopy (EIS), were performed under dark and visible light conditions using NaCl as the electrolyte. Moreover, the photoantibacterial/antibacterial properties of S−$CeO_2$ and (0.5, 1, 3 and 5%) Pd−$CeO_2$ NPs were also carried out for 24 h using bacterial strain *Staphylococcus aureus* (*S. aureus*) KCTC 1916 (Gram-positive) and *Pseudomonas aeruginosa* (*P. aeruginosa*) KCTC 1637 (Gram-negative).

## 2. Results and Discussion

### 2.1. Structural Analysis of CeO₂ and Pd−CeO₂ NPs

XRD provides a broad range of information related to the crystallographic nature and chemical structure of materials. The XRD patterns of the S−$CeO_2$ and Pd−$CeO_2$ NPs are

shown in Figure 1a, with diffraction peaks in the range of 20° to 80°. The peak positions at about 2θ = 28.56°, 33.27°, 47.36°, 56.40°, and 59.10° of each sample correspond to the (111), (200), (220), (311), and (222) planes of $CeO_2$ cubic fluorite phase (JCPDS no. 00-004-0593) (Figure S1). In $S-CeO_2$, the diffraction peak of $Ce_2O_3$ was observed, suggesting that mixed-phase $CeO_2/Ce_2O_3$ (JCPDS no. 00-023-1048) was present. It was found that gradually, the peak corresponding to $Ce_2O_3$ decreased with higher Pd doping. Hence, the peak intensity at 28.56° was seen to increase. This suggests that Pd doping inhibited the formation of $Ce_2O_3$ and suggested that the incorporation of $Pd^{2+}$ lowered the appearance of the $Ce_2O_3$ phase. The ionic radii of $Pd^{2+}$ and $Ce^{4+}$ are 86 and 101 pm, respectively. No other peaks were observed that contribute to the presence of PdO. Therefore, it can be concluded that $Pd^{2+}$ was successfully incorporated into the $CeO_2$ lattice.

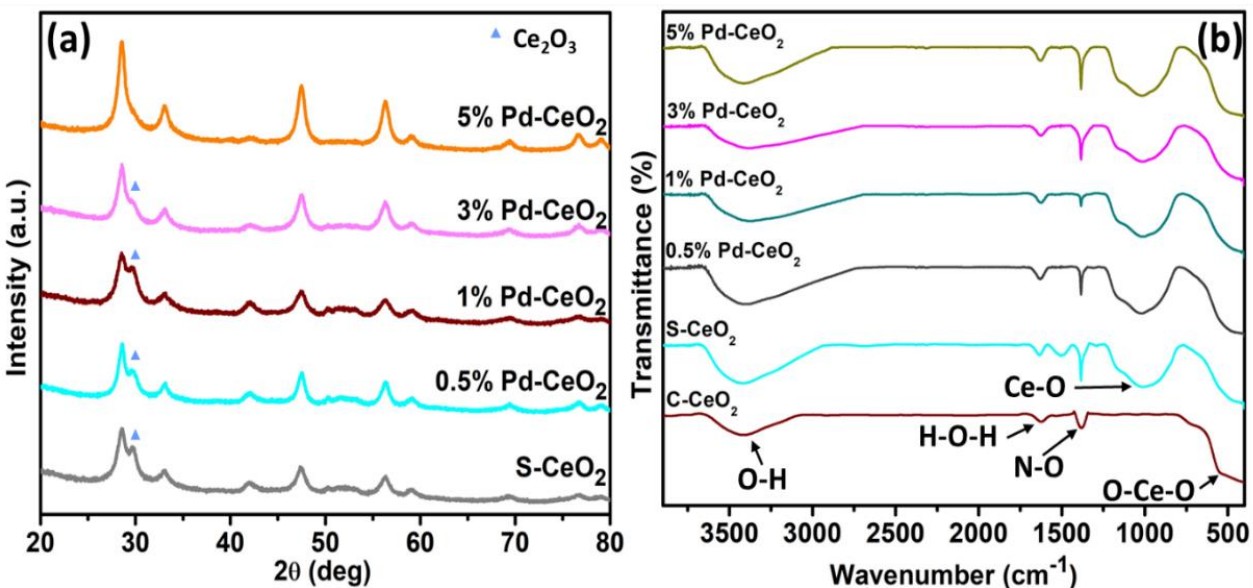

**Figure 1.** (**a**) XRD patterns and (**b**) FTIR-spectra of $S-CeO_2$ and (0.5, 1, 3, and 5%) $Pd-CeO_2$ NPs.

Furthermore, the average crystallite sizes were estimated using Debye-Scherrer's Formula (1):

$$D = \frac{0.9\lambda}{\beta cos\theta} \tag{1}$$

where $\beta$ is the full width at half maximum (FWHM), in radians, of the peak for a given (*hkl*) value, $\lambda$ = 1.5406 Å for the CuK$\alpha$ radiation used, and $\theta$ is the diffracting angle. The average crystallite sizes were found to be 34.67, 37.25, 16.25, 23.03, and 19.17 nm for $S-CeO_2$, 0.5%, 1%, 3%, and 5% $Pd-CeO_2$ NPs, respectively (Table 1).

**Table 1.** The average crystallite size (D), lattice parameter, cell volume, and average lattice strain of $S-CeO_2$ and $Pd-CeO_2$ NPs.

| Sample | Average Crystallite Size (D, nm) | Lattice Parameter, *a* (Å) | Cell Volume (Å³) | Average Lattice Strain |
|---|---|---|---|---|
| $S-CeO_2$ | 34.67 | 5.414 | 158.69 | 0.0022 |
| 0.5% $Pd-CeO_2$ | 37.25 | 5.408 | 158.16 | 0.0009 |
| 1% $Pd-CeO_2$ | 16.25 | 5.423 | 159.48 | 0.0004 |
| 3% $Pd-CeO_2$ | 23.03 | 5.409 | 158.25 | 0.0019 |
| 5% $Pd-CeO_2$ | 19.17 | 5.428 | 159.93 | 0.0022 |

The average lattice strain was also calculated using Equation (2):

$$\varepsilon = \frac{\beta_{hkl}}{4tan\theta} \tag{2}$$

The lattice parameters of $S-CeO_2$ and $Pd-CeO_2$ were observed to be comparable with the bulk $CeO_2$ (5.411 Å). The average lattice strain was decreased significantly when 0.5% and 1% Pd were incorporated. However, as more $Pd^{2+}$ were incorporated into the lattice, it would cause higher lattice strain. This can also be seen that as more Pd was doped, the cell volume increased slightly.

## 2.2. Fourier Transform-Infrared Spectroscopy of $CeO_2$ and $Pd-CeO_2$ NPs

FT-IR spectroscopic studies of $S-CeO_2$ and $Pd-CeO_2$ NPs were carried out within the range of 450–4000 $cm^{-1}$ at room temperature (Figure 1b). The broad band at approximately 3400 $cm^{-1}$ for all samples is attributed to the stretching mode of absorbed O-H in the samples. A broad, intense peak at 450 $cm^{-1}$ corresponds to the O-Ce-O stretching mode. At approximately 900 $cm^{-1}$, a broad peak was observed in $S-CeO_2$ and $Pd-CeO_2$ NPs, which corresponds to the bending of the intercalated Ce-O [39]. Moreover, a peak at about 1730 $cm^{-1}$ was also observed in all samples, and it could be assigned to the H-O-H bending mode.

## 2.3. X-ray Photoelectron Spectroscopy

XPS was performed at room temperature to investigate the chemical state and the electronic structure of the elements in $S-CeO_2$ and 0.5 and 5% $Pd-CeO_2$ NPs (Figure 2). Figure 2a shows the complete survey scan spectra of the samples, which confirmed the presence of Pd $3d$, O $1s$, and Ce $3d$. Figure 2b shows the six typical peaks for Ce $3d$. The peaks positioned at 883.43, 886.80, 896.26, 898.89, 905.59, and 914.75 eV is characteristic of $Ce^{4+}$ [40]. Peaks at 898.89, 905.59, and 914.75 eV correspond to $3d_{3/2}$, whereas peaks at 883.43, 886.80, and 896.26 eV correspond to $3d_{5/2}$ [41]. No obvious shift in the binding energy of Ce $3d$ spectra was observed.

Figure 2c shows the Pd $3d$ spectra of 0.5% $Pd-CeO_2$ and 5% $Pd-CeO_2$ NPs. As a result of the spin-orbit coupling, the Pd $3d$ spectra are split into two peaks of $3d_{5/2}$ and $3d_{3/2}$ at 335.03:335.03 eV and 340.40:340.31 eV, respectively. The XPS spectrum of O $1s$ can be seen in Figure 2d, in which all peaks exhibit two asymmetrical peaks, indicating the presence of $O^{2-}$, $OH^-$, and $O^-$ at the surface of the nanostructures. The peak at higher binding energy, i.e., 531–532 eV, is attributed to $O^{2-}$ vacancies and adsorbed -OH or $H_2O$, while the peak at lower binding energy, 529 eV, is attributed to the metal-oxygen binding [42]. $S-CeO_2$ showed a dominant peak at 531–532 eV, suggesting that $O^{2-}$ vacancies or adsorbed -OH were mainly found in the sample. On the other hand, 0.5 and 5% $Pd-CeO_2$ showed a dominant peak at 529 eV [42]. The difference might be due to the formation of the $Ce_2O_3$ phase in $S-CeO_2$ and the gradual decrease of the $Ce_2O_3$ phase in both 0.5 and 5% $Pd-CeO_2$, as shown in XRD. The typical C $1s$ were observed in the spectra (Figure 2e), which were derived from the carbon coating used in the analysis. The atomic concentration of C $1s$, O $1s$, Ce $3d$, and Pd $2p$ can be found in Table 2. The estimated Pd content in both 0.5% and 5% $Pd-CeO_2$ is lower than the expected value.

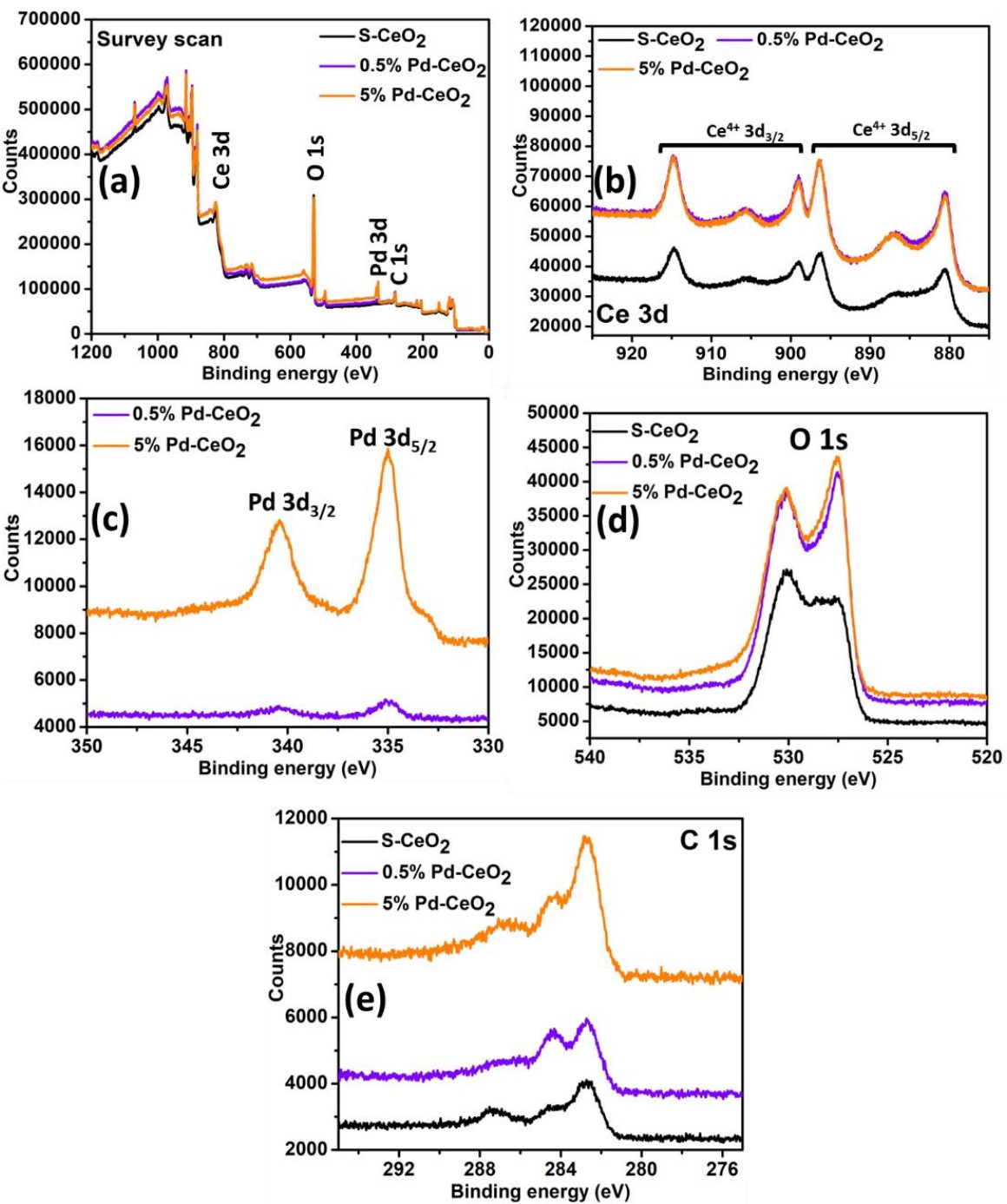

**Figure 2.** XPS spectra of S−CeO₂, 0.5% Pd−CeO₂, and 5% Pd−CeO₂: (**a**) Survey scan, (**b**) Ce 3*d*, (**c**) Pd 3*d*, (**d**) O 1*s*, and (**e**) C 1*s*.

**Table 2.** Atomic concentration of C 1*s*, O 1*s*, Ce 3*d*, and Pd 3*d* of S−CeO₂, 0.5% Pd−CeO₂, and 5% Pd−CeO₂.

| Sample | Atomic Concentration (%) | | | |
|---|---|---|---|---|
| | C 1*s* | O 1*s* | Ce 3*d* | Pd 3*d* |
| **S−CeO₂** | 12.4 | 74.1 | 13.5 | - |
| **0.5% Pd−CeO₂** | 12.6 | 72.1 | 15.1 | 0.2 |
| **5% Pd−CeO₂** | 19.0 | 66.3 | 12.8 | 1.9 |

The band gap reduction of S−CeO$_2$ and Pd−CeO$_2$ NPs might be due to the development of mid-gap states either above the valence band (VB) or below the conduction band (CB). Therefore, to study the cause of the reduction of band gap energy, VB-XPS was carried out for S−CeO$_2$ and Pd−CeO$_2$ NPs (Figure 3a).

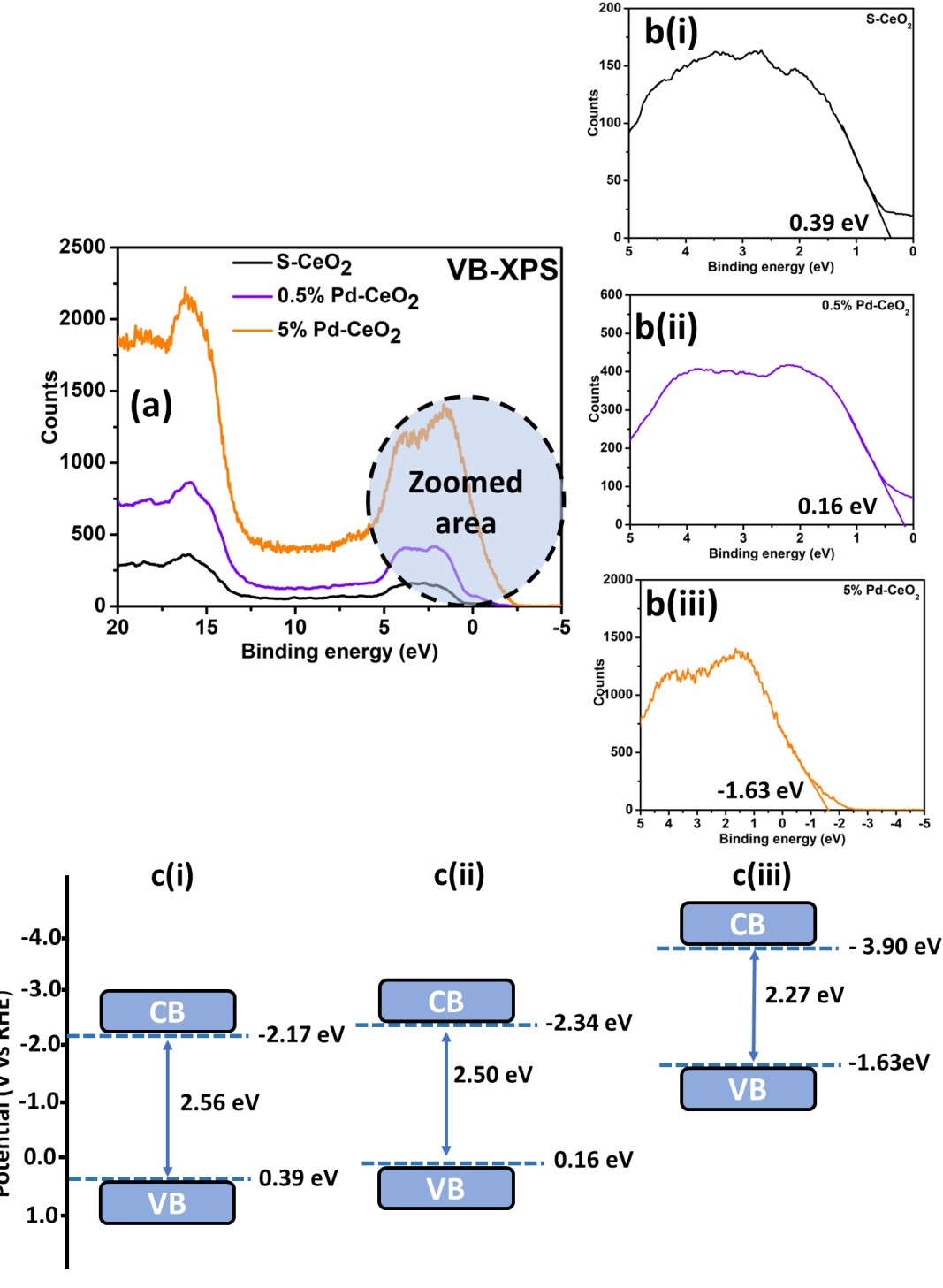

**Figure 3.** (**a**) Valence band XPS spectra of S−CeO$_2$, 0.5% Pd−CeO$_2$, and 5% Pd−CeO$_2$ NP; (**b**) zoomed valence band spectra of (**i**) S−CeO$_2$, (**ii**) 0.5% Pd−CeO$_2$, and (**iii**) 5% Pd−CeO$_2$ to estimate the band gap; and (**c**) proposed density of electronic states (DOS) for (**i**) S−CeO$_2$, (**ii**) 0.5% Pd−CeO$_2$, and (**iii**) 5% Pd−CeO$_2$ NPs.

The zoomed area of VB-XPS is shown in Figure 3b. The VB maximum of S−CeO$_2$, 0.5% Pd−CeO$_2$, and 5% Pd−CeO$_2$ NPs were found to be at 0.39, 0.16, and −1.63 eV, respectively (Figure 3b(i–iii)). The optical band gap energy obtained from the Tauc plot was 2.56, 2.50, and 2.27 eV for S−CeO$_2$, 0.5% Pd−CeO$_2$, and 5% Pd−CeO$_2$ NPs, respectively. Consequently, based on the materials' optical band gap and VB maxima, the CB minima would follow at −2.17, −2.34, and −3.90 eV for S−CeO$_2$, 0.5% Pd−CeO$_2$, and 5% Pd−CeO$_2$ NPs, respectively. The shifts of VB and CB result in the band gap energy reduction, as proposed in Figure 3c.

### 2.4. Optical Studies of CeO$_2$ and Pd−CeO$_2$ Using UV-vis-DRS and Photoluminescence Spectroscopy

UV-Vis DRS analysis is used to estimate the band gap energies of S−CeO$_2$ and Pd−CeO$_2$ NPs. The Tauc plot derived from the Kubelka–Munk function is shown in Figure 4a. The band gap energies of all samples were estimated using the Kubelka–Munk Equation (3):

$$\mathrm{F}(R) = \left( \frac{(1-R)^2}{2R} \times h\nu \right)^{\frac{1}{2}} \tag{3}$$

where *R* is the measured absolute reflectance of the samples. The band gap was obtained from the plots of $[\mathrm{F}(R)h\nu]^{1/2}$ versus $h\nu$, as the intercept of the extrapolated linear part of the plot at $[\mathrm{F}(R)h\nu]^{1/2} = 0$, assuming that the absorption coefficient ($\alpha$) is proportional to the Kubelka-Munk function F(*R*).

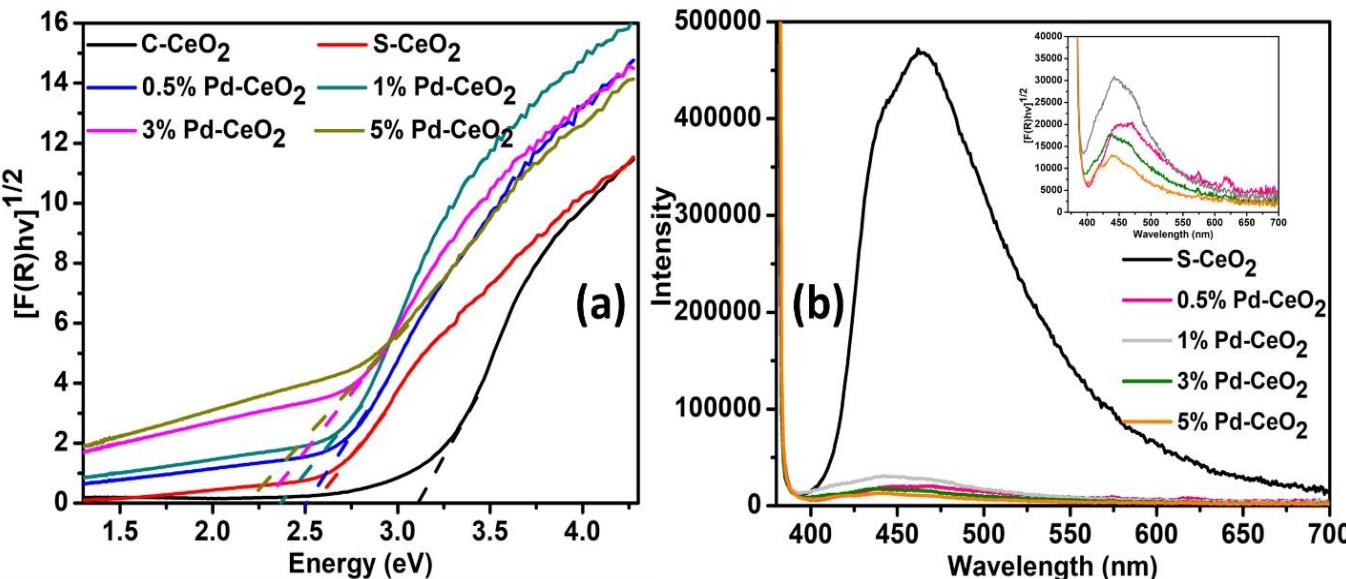

**Figure 4.** (**a**) Tauc plot derived from Kubelka-Munk function for band gap energy estimation, and (**b**) photoluminescence spectra of C−CeO$_2$, S−CeO$_2$, and Pd−CeO$_2$ NPs.

The band gap energy of C−CeO$_2$, S−CeO$_2$, 0.5% Pd−CeO$_2$, 1% Pd−CeO$_2$, 3% Pd−CeO$_2$, and 5% Pd−CeO$_2$ NPs were found to be decreasing (3.10 to 2.27 eV) as shown in Table 3. This shows that Pd doping is proven to decrease the band gap energy of the CeO$_2$. Pd doping is reported to help in reducing the energy difference between a CB and VB of CeO$_2$, which enhances the electronic conductivity of CeO$_2$. Moreover, oxygen vacancies (V$_o$) might be produced at the interface between the grains of Pd−CeO$_2$ NPs. Furthermore, the absorption edge of C−CeO$_2$, S−CeO$_2$, 0.5% Pd−CeO$_2$, 1% Pd−CeO$_2$, 3% Pd−CeO$_2$, and 5% Pd−CeO$_2$ NPs were greatly shifted into the visible region with the Pd doping as shown in Figure 4a.

**Table 3.** Band gap energy of $C-CeO_2$, $S-CeO_2$, and $Pd-CeO_2$ NPs.

| Sample | Band Gap Energy (eV) |
|:---:|:---:|
| $C-CeO_2$ | 3.10 |
| $S-CeO_2$ | 2.56 |
| 0.5% $Pd-CeO_2$ | 2.50 |
| 1% $Pd-CeO_2$ | 2.47 |
| 3% $Pd-CeO_2$ | 2.42 |
| 5% $Pd-CeO_2$ | 2.27 |

PL is used to evaluate the optical study of the crystals, defects on the surface and excitation fine structure of the semi-conducting materials. PL intensity describes the lifetime of electron relaxation from the VB to the CB, as well as the separation efficiency of photogenerated $e^-/h^+$ [43]. From Figure 4b, the PL intensity was decreased with more Pd doping concentration. This indicates that Pd doping can efficiently inhibit the $e^-/h^+$ recombination process due to the electron or hole trap levels (in this case, $Pd^{2+}/Pd^{3+}$ or $Pd^+/Pd^{2+}$) in the band structure of $CeO_2$. Moreover, the reduction in peak intensity might be due to the formation of more oxygen defects, increased surface defects, as well as maximum separation of charge carriers [44].

*2.5. Transmission Electron Microscopy Analysis of $CeO_2$ and $Pd-CeO_2$ NPs*

Figure 5 exhibits the TEM, HR-TEM, and SAED images of $S-CeO_2$, 0.5%, and 5% $Pd-CeO_2$ NPs. Both $S-CeO_2$ and $Pd-CeO_2$ (Figure 5a,d and g) have almost spherical morphology. At the same time, a rod-like structure was observed for $S-CeO_2$ NPs only. The average particle sizes of $S-CeO_2$, 0.5% $Pd-CeO_2$, and 5% $Pd-CeO_2$ are about 13, 9, and 6 nm, respectively. This shows that the incorporation of Pd into the lattice of $CeO_2$ decreased the particle size. The *d*-spacing value of the lattice planes was also determined from the HR-TEM images (Figure 5b,e,h). The *d*-spacing values for $S-CeO_2$, 0.5% $Pd-CeO_2$, and 5% $Pd-CeO_2$ NPs are estimated to be around 0.3, 0.2 and 0.1 nm, which corresponds to the (111), (200), and (220) planes of a fluorite-structure of cubic $CeO_2$ [45]. This is in accordance with the XRD analysis (Section 2.1).

Moreover, the nano-crystallinity of $S-CeO_2$ and $Pd-CeO_2$ NPs were examined by selected area electron diffraction (SAED) analysis, as shown in Figure 5c,f and i. It was observed that $S-CeO_2$, 0.5%, and 5% $Pd-CeO_2$ exhibit four prominent broad rings, which can be attributed to the (111), (200), (220), and (311) reflections of the fluorite cubic $CeO_2$ structure. This observation is supported by the XRD patterns reported earlier (Section 2.1).

The EDX mapping images shown in Figure S2a–c confirmed the presence of Ce, O, and Pd in $S-CeO_2$, 0.5%, and 5% $Pd-CeO_2$ NPs, respectively. The percentage of Ce was seen to decrease with more Pd content. The percentage mass of Ce, O, and Pd can be found in Table S1. This shows that Pd has been successfully incorporated into the $CeO_2$ lattice.

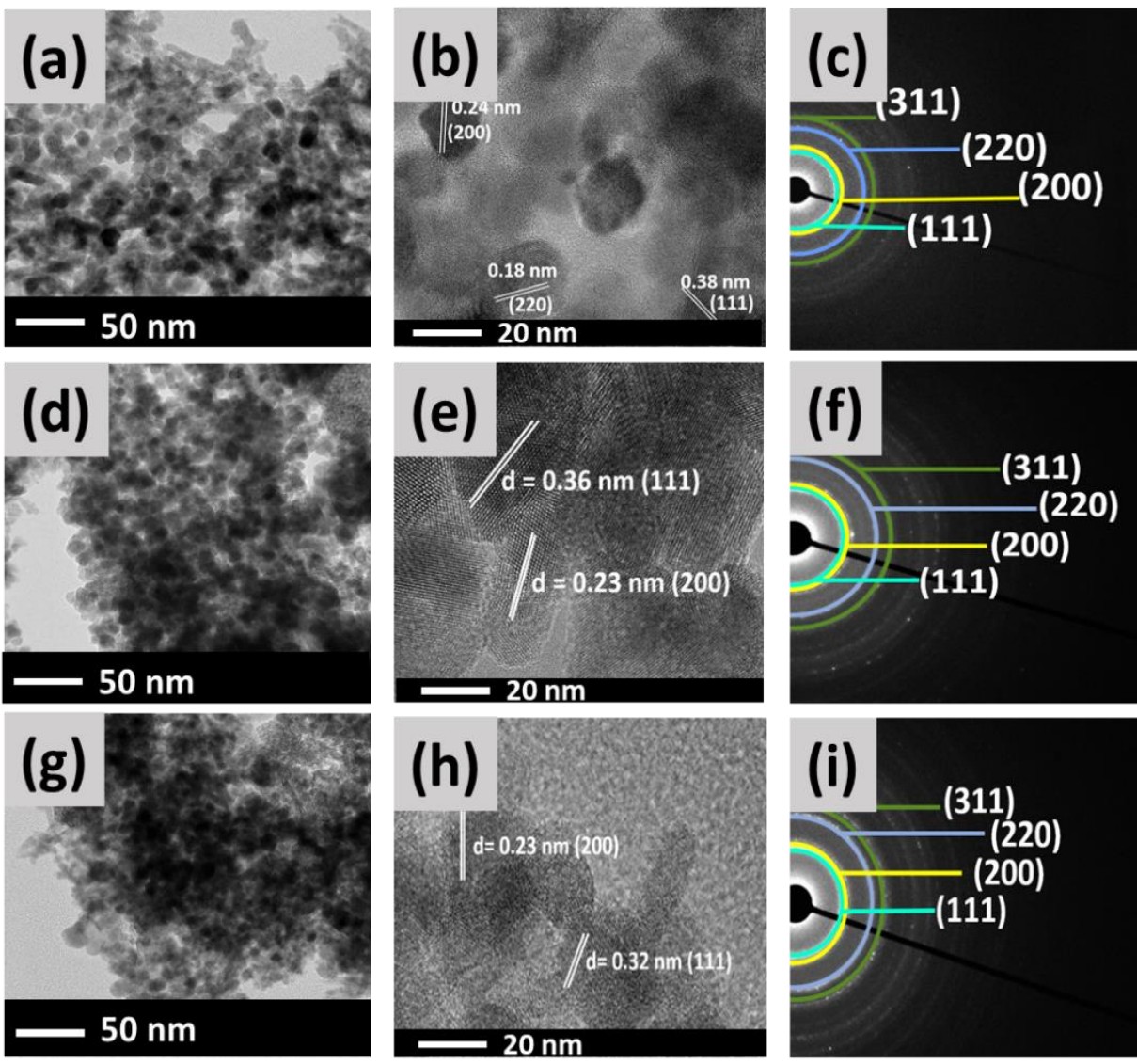

**Figure 5.** TEM, HR-TEM, and SAED patterns of (**a–c**) S−CeO₂, (**d–f**) 0.5% Pd−CeO₂, and (**g–i**) 5% Pd−CeO₂ NPs.

### 3. Applications

#### 3.1. Photocatalytic Degradation of MB

The photocatalytic degradation of MB using S−CeO₂ and Pd−CeO₂ NPs was carried out under visible light irradiation in a total experiment duration of 5 h. The activity of C−CeO₂ against MB dye under visible light was also investigated to compare the photocatalytic efficiency with the synthesized materials. Figure 6a shows the average $C/C_0$ of the photocatalytic degradation of MB using C−CeO₂, S−CeO₂, and Pd−CeO₂ NPs, while the average percentage of the photocatalytic degradation of MB and the absorption spectra of MB within 5 h can be seen in Figures S3 and S4, respectively. Based on the study, adsorption-desorption equilibrium was reached in 30 min, in which C−CeO₂ showed a slight adsorption capacity. About 39–61% adsorption was reached for the synthesized materials suggesting that the addition of Pd effectively increases the affinity of MB molecules towards photocatalysts [46]. The enhanced adsorption is also driven by the increase in the surface area contributed by significantly reduced particle size [46].

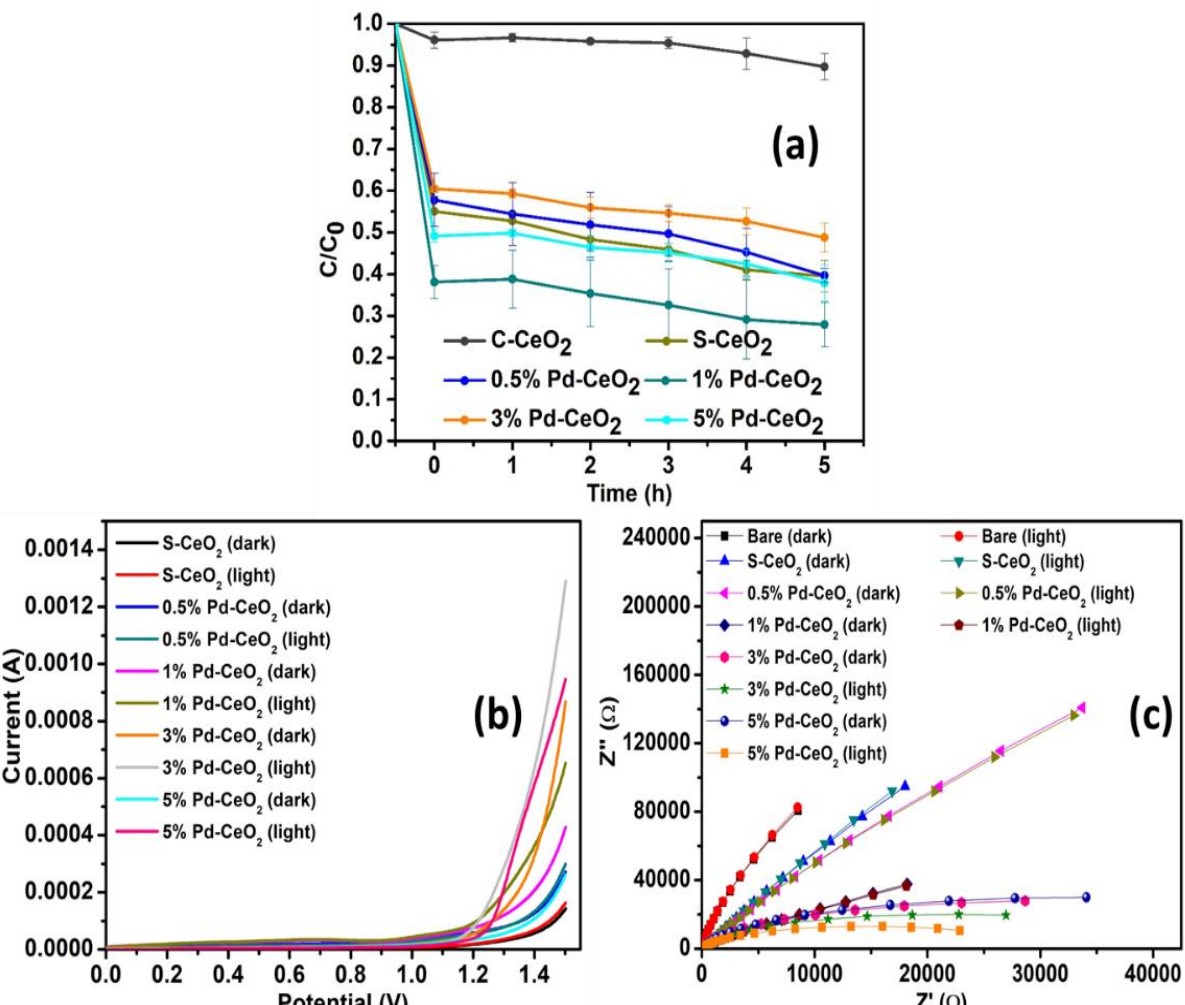

**Figure 6.** (**a**) The average $C/C_0$ of photocatalytic degradation of MB using $C-CeO_2$, $S-CeO_2$, and $Pd-CeO_2$ NPs under visible light irradiation, (**b**) LSV, and (**c**) EIS Nyquist plots of $S-CeO_2$, 0.5%, 1%, 3%, and 5% $Pd-CeO_2$ photoelectrode in the dark and under visible light irradiation.

The effectiveness of the photocatalysts in the photocatalytic degradation of MB was seen as follows: 1% $Pd-CeO_2$ > 5% $Pd-CeO_2$ > $S-CeO_2$ > 0.5% $Pd-CeO_2$ > 3% $Pd-CeO_2$ > $C-CeO_2$. A small response from $C-CeO_2$ might be due to the larger band gap energy, which was found to be 3.10 eV. Moreover, 3% $Pd-CeO_2$ and 0.5% $Pd-CeO_2$ also showed lower responses despite their lower band gap energies. This might be due to variations in particle size, which resulted in lower responses in the photocatalytic activity. In addition to that, a smaller surface area might also lead to less active sites present on the surface of the photocatalysts [47]. $S-CeO_2$ and 5% $Pd-CeO_2$ showed almost similar activities. The activity of $S-CeO_2$ was higher than some of the materials, which might be due to the mixed $CeO_2/Ce_2O_3$ phases, as shown in Figure 1a. The presence of both phases can ease the redox reaction between $Ce^{3+}$ and $Ce^{4+}$, which causes the photocatalytic activities to increase [48–50]. Choudhury et al. also stated that an electron from $Ce^{3+}$ can be transferred to adsorbed oxygen to form reactive oxygen species (ROS), i.e., superoxide radicals ($O_2^{\bullet-}$), whereas holes interact with water to form hydroxyl radicals ($^{\bullet}OH$) [51]. In addition, the $e^-$ in the CB can be trapped by oxygen vacancies and then react with $O_2$ to form $O_2^{\bullet-}$. These radicals are involved in the degradation of MB into harmless products [52]. Recombination of photogenerated $e^-$ and $h^+$ was restrained through the $Ce^{3+}/Ce^{4+}$ redox and oxygen vacancies which finally improved the photocatalytic performance of the $Pd-CeO_2$ materials [53]. Moreover, smaller particle size as well as low band gap energy also effec-

tively increased the photocatalytic activity [54]. Nevertheless, 5% Pd−CeO$_2$ showed a slightly better performance than S−CeO$_2$ due to its smaller particle size and band gap energy. Accordingly, 1% Pd−CeO$_2$ showed the highest response, which might be due to the optimum Pd loading. This shows that a further increase in doping does not enhance the photocatalytic degradation of MB [46]. Table 4 summarizes the average percentage of photocatalytic degradation of MB activities using S−CeO$_2$ and Pd−CeO$_2$ NPs under visible light irradiation.

**Table 4.** The average percentage MB dye degradation using C−CeO$_2$, S−CeO$_2$, and Pd−CeO$_2$ under visible light irradiation.

|  | **C−CeO$_2$** | **S−CeO$_2$** | **0.5% Pd−CeO$_2$** | **1% Pd−CeO$_2$** | **3% Pd−CeO$_2$** | **5% Pd−CeO$_2$** |
|---|---|---|---|---|---|---|
| **0 h** | 3.87 ± 1.94 | 44.92 ± 3.60 | 42.21 ± 6.34 | 61.87 ± 3.97 | 39.57 ± 0.81 | 50.83 ± 1.52 |
| **1 h** | 3.33 ± 1.04 | 47.22 ± 2.06 | 45.58 ± 7.54 | 61.20 ± 6.92 | 40.67 ± 1.03 | 50.18 ± 0.80 |
| **2 h** | 4.19 ± 0.66 | 51.69 ± 2.77 | 48.19 ± 7.73 | 64.63 ± 7.92 | 44.07 ± 2.50 | 53.62 ± 2.72 |
| **3 h** | 4.57 ± 1.33 | 54.13 ± 1.52 | 50.35 ± 6.55 | 67.42 ± 8.64 | 45.36 ± 2.02 | 54.85 ± 2.15 |
| **4 h** | 7.12 ± 3.82 | 58.99 ± 2.29 | 54.72 ± 5.71 | 70.87 ± 9.48 | 47.31 ± 3.19 | 57.56 ± 3.31 |
| **5 h** | 10.28 ± 3.13 | 60.48 ± 3.76 | 60.33 ± 1.76 | 72.07 ± 5.34 | 51.20 ± 3.47 | 62.10 ± 4.27 |

### 3.2. Photoelectrochemical Studies of Pd−CeO$_2$ NPs

The electrochemical and photoelectrochemical studies of S−CeO$_2$ and Pd−CeO$_2$ NPs were analyzed using LSV and EIS. Figure 6b shows the LSV analysis carried out in the dark and under visible light at 100 mV/s in the range of −0.7–1.5 V. S−CeO$_2$ showed the lowest photocurrent, and 5% Pd−CeO$_2$ showed the highest photocurrent. As the doping percentage increases (Figure 6b), the photocurrent response also increases. Each material, respectively, showed a higher response under visible light irradiation as compared to their response in the dark, suggesting that the materials were light-responsive. The improvement in the photocurrent response in Pd−CeO$_2$ was attributed to the light absorption ability due to the creation of a mid-gap state which lowered the band gap energy [55]. The band gap energy of Pd−CeO$_2$ was narrowed with higher Pd doping, in which the valence electrons can be excited to the conduction band state by absorbing light. It is stated that the e$^-$/h$^+$ pairs will recombine unless they are separated quickly [56]. In general, a high photocurrent suggests that the material has a strong ability for the generation and transfer of the photoexcited charge carrier under light irradiation [57]. Therefore, in this case, 5% Pd−CeO$_2$ exhibits better electron-hole separation, and it might be excited easily by visible light.

EIS was performed in the dark and under visible light irradiation at 0.0 V with a frequency ranging from 1–10$^6$ Hz (Figure 6c). The EIS Nyquist plots show the charge transfer resistance and separation efficiency between the photogenerated electrons and holes [58]. In general, a small arc radius and low resistance indicate higher charge transfer efficiency [55]. S−CeO$_2$ showed the highest resistance, followed by 0.5% and 1% Pd−CeO$_2$ indicating the slow interfacial charge-transfer process. This response might be due to their high band gap energies. In each case, there was no significant decrease in the resistance under visible light, which suggests that the transfer efficiency of the photogenerated electrons and holes was not accelerated under light irradiation. On the other hand, 3% Pd−CeO$_2$ and 5% Pd−CeO$_2$ showed significant responses, especially under visible light irradiation. This suggests that lower resistances derived from both 3% and 5% Pd−CeO$_2$ show a faster interfacial charge-transfer process. Other factors can also facilitate charge separation and transfer efficiency, such as the presence of Ce$^{3+}$ and oxygen vacancies [55].

### 3.3. Antibacterial Activities of Pd−CeO$_2$ NPs

The effect of bactericidal activities of S−CeO$_2$ NPs and Pd−CeO$_2$ NPs was examined against one Gram-positive bacteria (i.e., *S. aureus*) and another against Gram−negative pathogenic bacteria (i.e., *P. aeruginosa*). Since the synthesized NPs are light sensitive, the antibacterial activities were performed in the absence and presence of light. The

antibacterial effect of Pd−CeO$_2$ and S−CeO$_2$ NPs under light and dark conditions is shown in Figure 7. The results showed that both S−CeO$_2$ NPs and Pd−CeO$_2$NPs exhibit concentration-dependent inhibitions of *S. aureus* growth under light or dark conditions (Figure 7a,b). At a high concentration of 2048 µg/mL in the dark, the growth inhibition of *S. aureus* cells was found to be higher (reduction of 1.3 log CFU) in the presence of 5% Pd−CeO$_2$ NPs as compared to the CeO$_2$ NPs (reduction of 1.2 log CFU) (Figure 7a). Similarly, in the presence of light at 2048 µg/mL, the antibacterial effect of 5% Pd−CeO$_2$ NPs was also found to be higher (reduction of 1.6 log CFU) as compared to the S−CeO$_2$ NPs (reductions 1.0 log CFU) (Figure 7b). The bactericidal effect of CeO$_2$ NPs was found to be higher in the presence of light as compared to dark conditions. This finding suggests that light enhances the bactericidal effect of CeO$_2$ NPs towards Gram-positive bacteria *S. aureus*.

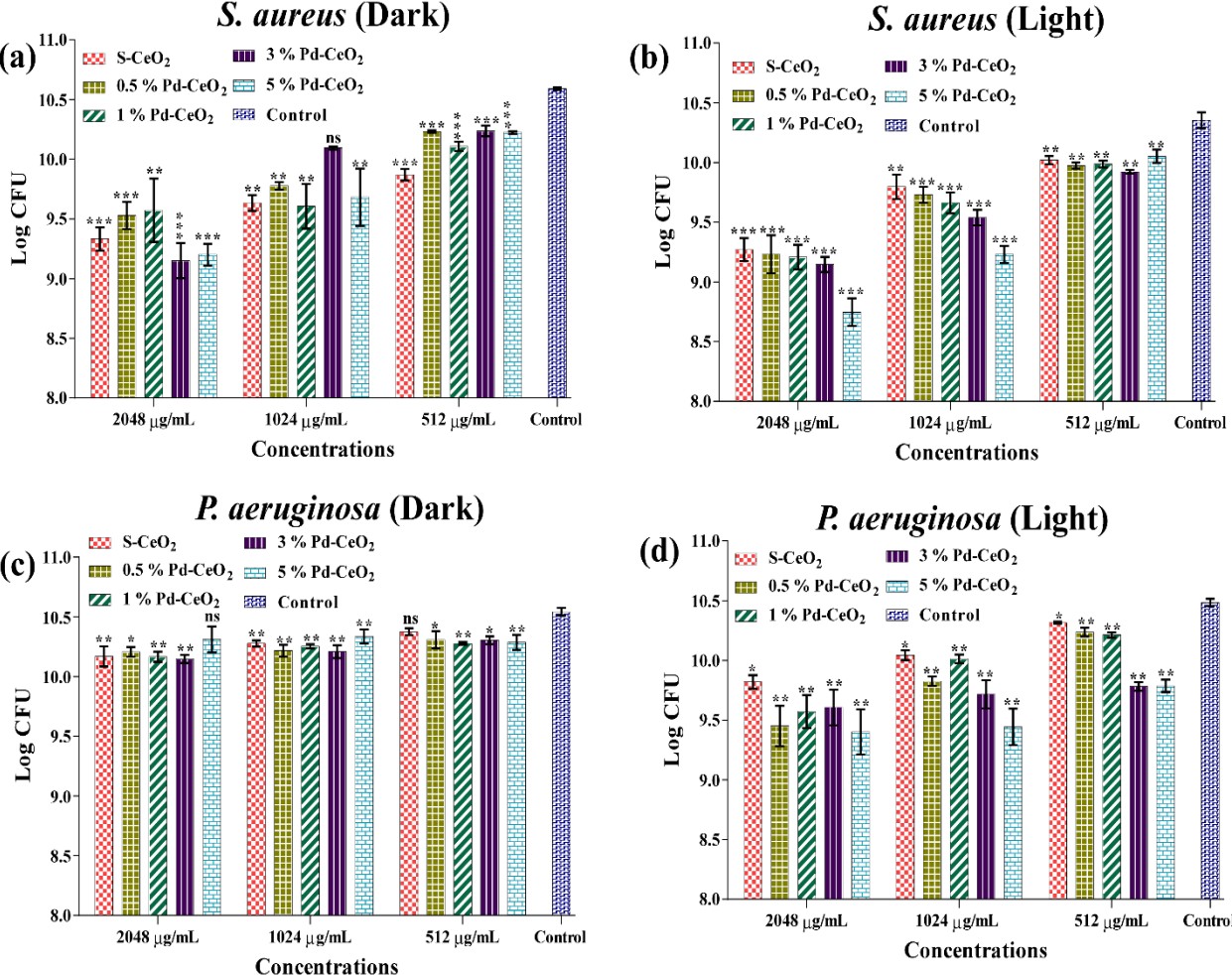

**Figure 7.** Log reduction of the *S. aureus* treated with different concentrations of S−CeO$_2$ NPs and Pd−CeO$_2$ NPs incubated under (**a**) dark and (**b**) light conditions. Log reduction of the *P. aeruginosa* treated with different concentrations of S−CeO$_2$ NPs and Pd−CeO$_2$ NPs incubated under (**c**) dark conditions and (**d**) light conditions, where ***, **, and * imply significance at $p < 0.0001$, $p < 0.01$, and $p < 0.05$, respectively, while ns means non-significant.

The antibacterial effect of S−CeO$_2$ NPs and Pd−CeO$_2$ NPs on *P. aeruginosa* was found to be different as compared to *S. aureus* (Figure 7c,d). There was no concentration-dependent inhibition of *P. aeruginosa* cell growth in the presence of light and dark conditions. The growth inhibition at the higher concentration (2048 µg/mL) and dark conditions was found to be non-significant in the presence of 5% Pd−CeO$_2$ NPs as compared to the S−CeO$_2$ NPs

(Figure 7c). The results in the presence of light were found to be significantly effective at all tested concentrations of S−CeO$_2$ NPs and Pd−CeO$_2$ NPs (Figure 7d).

In the presence of light and high concentration (2048 µg/mL), the 5% Pd−CeO$_2$ NPs showed a significant reduction of *P. aeruginosa* cells (1.0 log CFU) compared to the S−CeO$_2$ NPs (0.6 log CFU). Based on the findings of the present study, it is clear that light acts as a sensitizing agent, which results in the enhanced antibacterial activity of CeO$_2$ NPs as compared to dark conditions. Previous research has shown that UV light irradiation causes the formation of reactive oxygen species (ROS) such as $^\bullet$OH, $O_2^{\bullet-}$, and singlet oxygen (O$^-$) from metal-NPs [59–62]. This evidence can be explained in the current study by the possibility of ROS generation during the photocatalytic degradation of the MB experiment (Section 3.1). In addition, Pd-doping will also result in the enhanced antibacterial effect of CeO$_2$ NPs, and the effect was increased with the increasing concentration of Pd. Overall, these effects were found to be more effective towards Gram-positive bacteria *S. aureus* than the Gram-negative pathogen *P. aeruginosa*.

In general, the antibacterial activity of metal oxides is associated with the release of metal ions from metal oxides. It is reported that metal ions are involved in the destruction of the bacterial cell and membrane [63]. This could lead to the possibility of metal oxides penetrating the cell. In the case of CeO$_2$, CeO$_2$ dissociates into Ce$^{4+}$ ions (most of the time under irradiation of visible light) and interacts with the bacterial cell and penetrates the cell [64]. This might lead to changes in cell integrity, lactate dehydrogenase leakage and cell death. Moreover, the generation of ROS is also responsible for antibacterial activity [65]. When CeO$_2$ is irradiated with light, e$^-$ are promoted to CB and reacted with molecular oxygen to form $O_2^{\bullet-}$. The photogenerated h$^+$ reacts with H$_2$O molecules on the surface of CeO$_2$ to form $^\bullet$OH radicals. These two radicals are believed to be contributing significantly to the destruction of the bacterial cell membrane [60].

## 4. Experimental

### 4.1. Chemicals, Bacterial Strain, Culture Media, and Growth Conditions

Cerium(III) nitrate hexahydrate 98% (Ce(NO$_3$)$_3$·6H$_2$O) and commercial CeO$_2$ (C−CeO$_2$) were purchased from Sigma-Aldrich whereas, palladium(II) chloride anhydrous (PdCl$_2$) was obtained from Fluka. Water was purified using Aquatron (England) prior to use. For photocatalytic degradation of MB and photoelectrochemical studies, MB and NaCl were obtained from Merck. For electrode preparation, ethanol (95%), ethyl cellulose 48–49.5% (*w/w*) ethoxy basis, and alpha-terpineol were obtained from Daejung Chemicals and Metals Co., Ltd., Sigma-Aldrich and Merck, respectively. The bacterial strain *S. aureus* KCTC 1916 (Gram-positive) and *P. aeruginosa* KCTC 1637 (Gram-negative) were purchased from the Korean Collection for Type Cultures (KCTC, Daejeon, Korea). The bacteria were cultivated using Tryptic soy broth (TSB) and a TSB agar plate (Difco Laboratory Inc., Detroit, MI, USA). The growth temperature of the bacterial strain was 37 °C under aerobic conditions.

### 4.2. Instrumentations

CeO$_2$ and (0.5, 1, 3, and 5%) Pd−doped CeO$_2$ NPs were synthesized using a microwave-assisted method (Anton Paar Monowave 400, Graz, Austria). Fourier Transform-Infrared Spectroscopy (FT-IR) was used to identify the possible functional groups present in the synthesized CeO$_2$ and Pd−CeO$_2$ NPs using Shimadzu IRPrestige-21 FT-IR Spectropho-tometer. The stretching frequencies obtained were plotted as %transmittance mode on the y-axis and wavenumber (cm$^{-1}$) on the x-axis from 450 to 4000 cm$^{-1}$. The morphology, structure, and elemental mapping were analyzed using field emission transmission electron microscopy (FE-TEM) and selected area electron diffraction (SAED) using JEM-F200 (JEOL Ltd., Tokyo, Japan). The determination of band gap energy of S−CeO$_2$ and Pd−CeO$_2$ NPs was investigated using UV-Vis diffuse reflectance spectroscopy (DRS) (Shimadzu, UV-2600, Tokyo, Japan). A photoluminescence (PL) study was carried out using F-7000 Fluorescence spectroscopy (Hitachi High Tech, Tokyo, Japan) with an excitation wave-length of 370 nm. X-ray photoelectron spectroscopy (XPS) and valence band (VB)-XPS were

performed (Kratos Analytical, AXIS Nova, Manchester, UK). Photocatalytic activities of MB dye degradation were carried out using a photochemical reactor Toption (TOPT-V) having a 300 W Xenon lamp, and the photocatalytic degradation of MB was monitored using UV–visible spectrophotometer (Shimadzu UV-1900, Japan). Photoelectrochemical studies were performed using Autolab (MetroHm, Herisau, Switzerland) under dark and visible light irradiation (Simon FL30 LED Floodlight, 100 W, Jiangsu, China). The measurements were carried out and taken from NOVA software. Photoantibacterial studies of $S-CeO_2$ and $Pd-CeO_2$ NPs were investigated using a 96-well microplate in the presence of the LED light (One & One Plus OP-0303 Nape Slim LED Stand).

### 4.3. Microwave-Assisted Synthesis of Cerium Oxide Nanoparticles

$CeO_2$ NPs were synthesized using the microwave-assisted method. Briefly, 0.05 M of $Ce(NO_3)_3 \cdot 6H_2O$ solution (15 mL) was prepared in a microwave vessel. Exactly 2.4 mL of 1 M NaOH was added dropwise into the solution. Subsequently, the microwave reaction was carried out at 180 °C for 15 min at 850 W microwave power. The precipitate formed was centrifuged and washed three times with water before it was dried at 80 °C. The product was stored and labeled as $S-CeO_2$.

### 4.4. Microwave-Assisted Synthesis of Pd−CeO₂ Nanoparticles

$Pd-CeO_2$ NPs were synthesized using the same method as mentioned above. A 15 mL of 0.05 M $Ce(NO_3)_3$ solution was prepared. A specific amount of $PdCl_2$ was then added to prepare 0.5, 1, 3, and 5% $Pd-CeO_2$. Subsequently, 1 M NaOH was added dropwise into the solution. The microwave reaction was carried out at 180 °C for 15 min at 850 W microwave power. The precipitate was formed, centrifuged, and washed three times with water before it was dried at 80 °C. The products were stored and labeled as 0.5% $Pd-CeO_2$, 1% $Pd-CeO_2$, 3% $Pd-CeO_2$, and 5% $Pd-CeO_2$ for further use.

### 4.5. Electrode Preparation

$S-CeO_2$, 0.5%, 1%, 3%, and 5% $Pd-CeO_2$ electrodes were prepared using the doctor blade method. In brief, 25 mg of the respective sample was mixed with 0.5 mL ethanol and 0.5 mL α-terpineol. The mixture was sonicated for 10 min. Then, 25 mg of ethyl cellulose was added to the mixture and stirred at 80 °C for ~2 h to produce a thick paste. The paste was then spread on fluorine-doped tin oxide (FTO) glass electrode by 2 cm × 1 cm using a doctor blade. The glass electrode was dried at 80 °C for 24 h prior to use.

### 4.6. Photocatalytic Degradation of MB Dye

The photocatalytic degradation of MB dye using $S-CeO_2$ and $Pd-CeO_2$ NPs under visible light irradiation was monitored using UV-Vis spectroscopy. Exactly 10 mg of $S-CeO_2$ and 0.5, 1, 3, and 5% $Pd-CeO_2$ NPs were dispersed in 50 mL of 10 ppm MB dye solution. The sample mixture was stirred in the dark for 30 min to reach adsorption-desorption equilibrium. Then, the reaction tubes were irradiated with visible light (300 W) for 5 h in which the absorbance of the solution was measured every 1 h. The percentage of photocatalytic dye degradation of MB was obtained using Equation (4):

$$\% \text{ photocatalytic MB dye degradation} = \frac{\left(A_{blank} - A_{sample}\right)}{A_{blank}} \times 100 \tag{4}$$

where $A_{blank}$ is the absorbance of MB only and $A_{sample}$ is the absorbance of MB dye after photocatalytic degradation by the photocatalyst.

### 4.7. Photoelectrochemical Studies

The photoelectrochemical response of $S-CeO_2$ and $Pd-CeO_2$ NPs was examined through linear sweep voltammetry (LSV) and electrochemical impedance spectroscopy (EIS) experiments. The experiments were carried out under ambient conditions in the dark

and under visible light irradiation (LED, 100 W) in 100 mL of a 1 M NaCl aqueous electrolyte solution. The distance between the light and reactor vial was 22 cm. The prepared glass electrode, Ag/AgCl electrode, and Pt electrode were used as the working electrode, reference electrode, and counter electrode, respectively. For each electrode, LSV and EIS were performed in the dark and later under visible light irradiation at 100 mV/s in the potential range of $-0.7$–1.5 V and at 0.0 V with a frequency ranging from $1$–$10^6$ Hz, respectively.

### 4.8. Assays for Antibacterial Activities of $Pd-CeO_2$ Nanoparticles

A single colony from the agar plate of each bacterium was inoculated in the TSB and incubated for 12 h at 37 °C under shaking conditions (250 rpm). The seed culture was diluted into fresh TSB in order to achieve the final $OD_{600}$ ~0.05. The diluted cell culture (300 μL) was placed in a 96-well microtiter plate and treated with different concentrations of $S-CeO_2$ and 0.5, 1, 3, and 5% $Pd-CeO_2$ NPs. The concentrations of these NPs ranged from 265 to 2048 μg/mL. Two sets of microtiter plates were prepared to contain different concentrations of NPs. One set of microplates was incubated in the dark for 24 h at 37 °C. The second set of the microplates was incubated in the presence of LED light that was set at the height of 27 cm for 24 h at 37 °C. After incubation, the cell culture (100 μL) was transferred into another 96-well microplate containing 200 μL sterile TSB. A two-fold serial dilution (up to $10^{-8}$ dilutions) of the cell culture was carried out. The serially diluted cell culture (100 μL) was spread-plated on the TSA plates. The TSA plates were incubated at 37 °C for 24 h, followed by the counting of the total colonies. The log colony forming unit (CFU) values of treated and untreated cells were calculated. All the experiments were carried out in triplicate.

### 5. Conclusions

$S-CeO_2$ and 0.5, 1, 3, and 5% $Pd-CeO_2$ NPs were successfully synthesized using the microwave-assisted method. Mixed phases of $CeO_2/Ce_2O_3$ were obtained in $S-CeO_2$ and 0.5, 1, and 3% $Pd-CeO_2$. However, there were no mixed phases observed in 5% $Pd-CeO_2$ NPs. Average crystallite sizes were found to be between 16.25–37.25 nm. Based on the TEM images, irregularly spherical shaped $S-CeO_2$ and $Pd-CeO_2$ NPs were observed with an average particle size between 6 and 13 nm. The band gap energy of $S-CeO_2$ decreased with the addition of higher Pd content, in which the band gap narrowing phenomena was illustrated through the DOS scheme. One percent of $Pd-CeO_2$ NPs showed enhanced responses under visible light irradiation in photocatalytic degradation of MB. Meanwhile, 5% $Pd-CeO_2$ showed enhanced responses in photoelectrochemical and photoantibacterial activities. Therefore, $Pd-CeO_2$ has shown visible light active properties and can be potentially used in photocatalysis and photoantibacterial applications.

**Supplementary Materials:** The following supporting information can be downloaded at: https://www.mdpi.com/article/10.3390/catal13010096/s1. Figure S1. Standards XRD patterns of $CeO_2$, $Ce_2O_3$, and PdO; Figure S2. EDX-mapping of Ce, O, and Pd elements in (a) $S-CeO_2$, (b) 0.5% $Pd-CeO_2$, and (c) 5% $Pd-CeO_2$ NPs; Figure S3. The average percentage of photocatalytic degradation of MB dye using $C-CeO_2$, $S-CeO_2$, and $Pd-CeO_2$ NPs under visible light irradiation; Figure S4. Absorbance spectra of photocatalytic degradation of MB using (a) $C-CeO_2$, (b) $S-CeO_2$ (c) 0.5%, (d) 1%, (e) 3%, and (f) 5% $Pd-CeO_2$ NPs; Table S1. Percentage mass of Ce, O, and Pd elements in $S-CeO_2$, 0.5% $Pd-CeO_2$, and 5% $Pd-CeO_2$ NPs using EDX-Mapping.

**Author Contributions:** S.N.M.: Methodology, Investigation, Data curation, and Writing—original draft. F.K.: Methodology, Investigation, Data curation, and Writing. M.H.H.: Supervision, Writing—review and editing. Y.-M.K.: Resources, Formal analysis. M.M.K.: Supervision, Conceptualization, Funding acquisition, Writing—review and editing. All authors have read and agreed to the published version of the manuscript.

**Funding:** The authors would like to acknowledge the FRC grant (UBD/RSCH/1.4/FICBF(b)/2022/046) received from Universiti Brunei Darussalam, Brunei Darussalam. This research was supported by

Basic Science Research Program through the National Research Foundation of Korea (NRF) grant funded by the Ministry of Education (2021R1A6A1A03039211 and 2022R1A2B5B01001998).

**Conflicts of Interest:** The authors declare no conflict of interest.

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
