# Peer review of "Effect of Pd-Doping Concentrations on the Photocatalytic, Photoelectrochemical, and Photoantibacterial Properties of CeO2"

_catalysts, doi:10.3390/catal13010096_

Round 1

Reviewer 1 Report

Comments to the authors

In the present manuscript the authors reported the enhancement of the photocatalytic, photoelectrochemical, and photoantibacterial properties of CeO2 by doping with Pd. Enhancement of the photocatlytic and antibacterial activities of metal oxides by doping with metals specially noble metals has received increased attention in the last decades. The manuscript is well written and the research work is supported with scientific references. The manuscript contain certain flaws and deficiency, therefore I suggest minor revision for quality enhancement, which are given bellow.

1.      The manuscript contains a large number of grammatical and typing mistakes which should be carefully read and remove and the language need sufficient improvement.

2.      Discuss catalytic and biological applications of CeO2 in the introduction section.

3.      Mention the toxic effect of methylene blue dye in the introduction section using and citing this article; Water 2022, 14, 242. https://doi.org/10.3390/w14020242.

4.       The authors should properly correlate the XRD data with the results presented in the manuscript.

5.      What is C-CeO2? Clarify it.

6.      Authors are suggested to explain the mechanism of photodegrdation of dye degradation by using and citing this article; Journal of Environmental Chemical Engineering 8 (2020) 104364.

7.      Explain the mechanism of the antibacterial activity.

8.      Mention the scientific results obtained in the abstract.

9.      Explain the novelty of your work as a lot of work are published on Pd-CeO2 e.g. Solid State Sciences, 87, 2019, 9-14.

10.  Put all the figures after their discussion.

11.  Check the sustainability of the synthesized material by recycling and reusing for dye degradation.

Author Response

Reviewer 1:

Comments to the authors

In the present manuscript the authors reported the enhancement of the photocatalytic, photoelectrochemical, and photoantibacterial properties of CeO2 by doping with Pd. Enhancement of the photocatlytic and antibacterial activities of metal oxides by doping with metals specially noble metals has received increased attention in the last decades. The manuscript is well written and the research work is supported with scientific references. The manuscript contain certain flaws and deficiency, therefore I suggest minor revision for quality enhancement, which are given bellow.

  1. The manuscript contains a large number of grammatical and typing mistakes which should be carefully read and remove and the language need sufficient improvement.

Reply: Thank you for your comments. We have rechecked the manuscript for any mistakes and improved the grammar.

  1. Discuss catalytic and biological applications of CeO2 in the introduction section.

Reply: Thank you for your suggestion. The discussion on antibacterial and dye degradation can be found on page 2 and 3, Section 1.0, 1st, 2nd, and 4th paragraphs.

  1. Mention the toxic effect of methylene blue dye in the introduction section using and citing this article; Water 2022, 14, 242. https://doi.org/10.3390/w14020242.

Reply: Thank you for your comment. The toxic effect of MB has been added on page 2, 2nd paragraph.

  1. The authors should properly correlate the XRD data with the results presented in the manuscript.

Reply: Thank you for your comment. The XRD data was correlated with photocatalysis results on page 16, 2nd paragraph.

  1. What is C-CeO2? Clarify it.

Reply: Thank you for your comment. C-CeO2 is commercially available CeO2 obtained from Sigma-aldrich. C-CeO2 has already been clarified on page 3, under section 2.1.

  1. Authors are suggested to explain the mechanism of photodegrdation of dye degradation by using and citing this article; Journal of Environmental Chemical Engineering 8 (2020) 104364.

Reply: Thank you for your suggestion. The mechanism of photocatalytic dye degradation has been explained on page 16, 2nd paragraph. The suggested article has been cited.

  1. Explain the mechanism of the antibacterial activity.

Reply: Thank you for your comments. The antibacterial mechanism has been added on page 19.

  1. Mention the scientific results obtained in the abstract.

Reply: Thank you for your comment. Some of the scientific results has been mentioned in the abstract.

  1. Explain the novelty of your work as a lot of work are published on Pd-CeO2 e.g. Solid State Sciences, 87, 2019, 9-14.

Reply: Thank you for your comment. The novelty of the work has been stated on page 3, 3rd paragraph.

  1. Put all the figures after their discussion.

Reply: Thank you for your suggestion. All figures have been moved to after each discussion.

  1. Check the sustainability of the synthesized material by recycling and reusing for dye degradation.

Reply: Thank you for your suggestion. However, due to limited resources, this experiment cannot be carried out.

Reviewer 2 Report

The authors reported the microwave-assisted synthesis of CeO2 and Palladium-doped CeO2 (Pd-CeO2) NPs. They found that the Ce2O3 phase gradually disappeared upon doping with a higher percentage of Pd. Besides, the doping of Pd induces a reduction in the band gap of S-CeO2 and Pd-CeO2 NP and the PL intensities. Moreover, they found Pd-CeO2 NPs showed enhanced activities in photocatalytic degradation of methylene blue, photoelectrochemical, and photoantibacterial activities. The results are interesting while the organization of data requires further improvement. Here are several questions and suggestions:

1. The abstract part lacks the research motivation. The authors are suggested to add 1-2 sentences about difficulties or problems in this field and explain why Pd-doping in CeO2 is valuable.

2. The authors claim in the introduction part that the Incorporation of Pd2+ into the CeO2 crystal lattice has shown better catalytic activity than the palladium-supported CeO2 catalyst. Since it is known that doping Pd in CeO2 enhances activity, what are the merits of this work? Please emphasize.

3. It is strongly suggested to polish the language and figures for better demonstration. For example, the panel labels in figure 3 are confusing. There are (a), b(i), b (ii), b(iii), and another (a), (b), (c).

4. In Figure 4b, the PL intensity decreased dramatically with only 0.5% Pd-doping and maintained with an increase in Pd concentration up to 5%. The authors attributed this to electron or hole trap levels. However, the bandgap decreased with increased Pd-doping concentration(Table 3), the recombination should be easier which results in enhanced PL intensity. How do authors explain these two sharply different effects?

Author Response

Reviewer 2:

Comments to the authors

The authors reported the microwave-assisted synthesis of CeO2 and Palladium-doped CeO2 (Pd-CeO2) NPs. They found that the Ce2O3 phase gradually disappeared upon doping with a higher percentage of Pd. Besides, the doping of Pd induces a reduction in the band gap of S-CeO2 and Pd-CeO2 NP and the PL intensities. Moreover, they found Pd-CeO2 NPs showed enhanced activities in photocatalytic degradation of methylene blue, photoelectrochemical, and photoantibacterial activities. The results are interesting while the organization of data requires further improvement. Here are several questions and suggestions:

  1. The abstract part lacks the research motivation. The authors are suggested to add 1-2 sentences about difficulties or problems in this field and explain why Pd-doping in CeO2 is valuable.

Reply: Thank you for your suggestion. The abstract has been revised and updated.

  1. The authors claim in the introduction part that the Incorporation of Pd2+ into the CeO2 crystal lattice has shown better catalytic activity than the palladium-supported CeO2 catalyst. Since it is known that doping Pd in CeO2 enhances activity, what are the merits of this work? Please emphasize.

Reply: Thank you for your comment. In this work, CeO2/Ce2O3 mixed phases were produced via microwave-assisted synthesis method. The doping of Pd has shown to influence the phase of CeO2 in which Ce2O3 phase has disappeared with more Pd doping. This finding has not been greatly explored and reported, according to the authors’ best knowledge. Therefore, this finding offers new idea to be explored in further studies.

  1. It is strongly suggested to polish the language and figures for better demonstration. For example, the panel labels in figure 3 are confusing. There are (a), b(i), b (ii), b(iii), and another (a), (b), (c).

Reply: Thank you for your comments. We have improved the grammar and the labels for Figure 3 have been revised.

  1. In Figure 4b, the PL intensity decreased dramatically with only 0.5% Pd-doping and maintained with an increase in Pd concentration up to 5%. The authors attributed this to electron or hole trap levels. However, the bandgap decreased with increased Pd-doping concentration(Table 3), the recombination should be easier which results in enhanced PL intensity. How do authors explain these two sharply different effects?

Reply: Thank you for your comment. Based on our discussion section, even though there is a decrease in the band gap upon Pd doping, the Pd doping also decreased the PL intensity. This indicates that the Pd inhibits the e-/h+ recombination as discussed on page 14, 1st paragraph and as shown Figure 4b.